# Deep Learning for Elucidating Modifications to RNA—Status and Challenges Ahead

**DOI:** 10.3390/genes15050629

**Published:** 2024-05-15

**Authors:** Sarah Rennie

**Affiliations:** Section for Computational and RNA Biology, Department of Biology, University of Copenhagen, 2200 Copenhagen, Denmark; sarah.rennie@bio.ku.dk

**Keywords:** RNA-binding proteins (RBPs), post-transcriptional modifications, deep learning, neural networks, sequence motifs

## Abstract

RNA-binding proteins and chemical modifications to RNA play vital roles in the co- and post-transcriptional regulation of genes. In order to fully decipher their biological roles, it is an essential task to catalogue their precise target locations along with their preferred contexts and sequence-based determinants. Recently, deep learning approaches have significantly advanced in this field. These methods can predict the presence or absence of modification at specific genomic regions based on diverse features, particularly sequence and secondary structure, allowing us to decipher the highly non-linear sequence patterns and structures that underlie site preferences. This article provides an overview of how deep learning is being applied to this area, with a particular focus on the problem of mRNA-RBP binding, while also considering other types of chemical modification to RNA. It discusses how different types of model can handle sequence-based and/or secondary-structure-based inputs, the process of model training, including choice of negative regions and separating sets for testing and training, and offers recommendations for developing biologically relevant models. Finally, it highlights four key areas that are crucial for advancing the field.

## 1. Introduction

Modifications to RNA play pivotal roles in regulating gene expression, influencing a wide range of processes such as cellular differentiation, development, stress response and disease pathogenesis [1,2]. The term “epitranscriptomics” refers to the study of a broad range of modifications that influence RNA either co- or post-transcriptionally. Notable examples of modifications that play vital roles in regulating the fate of RNAs of protein-coding genes include RNA-binding proteins (RBPs) and a host of chemical modifications such as N6-methyladenosine (m${\;\;}^{6}$A), m5-cytosine (m${\;\;}^{6}$A), pseudouridine (ψ), and adenosine-to-inosine (A-to-I) RNA editing (Figure 1A). These modifications exhibit diverse cellular functions, including mRNA stability or degradation, splicing regulation, translation, and transport and localisation of mRNA targets (Figure 1B) [3,4]. In humans, it is estimated that there are over 1500 RBPs, which divide into a number of sub-families and possess a range of RNA-binding domains, including the RNA recognition motif (RRM), K-homology domain (KH), double-stranded RNA-binding domain (dsRBD), and zinc-finger domains. Furthermore, RBPs tend to localise to specific sub-cellular compartments, exhibit context-specific binding patterns, and often have highly specific sequence preferences [5]. The most well-studied chemical modification, m${\;\;}^{6}$A, is typically found in the 3${\;\;}^{\prime }$ untranslated region (3${\;\;}^{\prime }$ UTR) and around the stop codon of modified transcripts [6], and its functions are mediated by a specific set of RBPs known as m${\;\;}^{6}$A readers [7]. In contrast to m${\;\;}^{6}$A, which is reversible, A-to-I RNA editing is an irreversible modification catalysed by the enzymatic activities of the RBP protein family Adenosine deaminases acting on RNA (ADAR), which act on double-stranded RNA (dsRNA) and play particularly important roles in self or non-self recognition in the regulation of immune response [8]. Whilst individual RNA modifications have profound consequences on RNA function, they do not behave independently, and how they coordinate to exert their regulatory effects is highly complex and has significant implications for understanding gene regulatory control. Before this complexity can be fully deciphered, however, a crucial prerequisite is to determine which transcripts and precise positions are targeted by each modification and in what contexts, together with their sequence or structural preferences for binding.

Modified locations within the transcriptome can be detected experimentally, most commonly through the use of immunoprecipitation-based methods [9,10,11]. For the detection of RBP binding sites in vivo, numerous variations of the standard protocol, cross-linking followed by immunoprecipitation and sequencing (CLIP), exist. Notable examples are the high-resolution iCLIP (individual-nucleotide resolution CLIP) [12], and eCLIP (enhanced CLIP) [13], the latter of which has been extensively applied by the ENCODE consortium to profile 150 RBPs in the K562 and HepG2 cell lines, forming the largest resource of RBP binding to date [14]. However, CLIP-based methods are susceptible to several biases, including preferential binding to specific RNA sequences and the efficiency of cross-linking can vary significantly between different RBPs and RNA regions [15]. This creates challenges for the accurate determination of binding sites, potentially leading to both false positives and false negatives in the collected data. Recent advancements have led to the development of immunoprecipitation-free methods for detecting RBP protein binding, such as RNA-editor methods like HyperTRIBE and DART-seq [16,17]. These methods provide promising alternatives for modification detection, circumventing some of the biases of CLIP methods, although they do not guarantee the detection of the exact binding location. In addition to in vivo methods, in vitro approaches such as RNAcompete, RNA Bind-n-Seq, and SELEX (Systematic Evolution of Ligands by EXponential enrichment) facilitate the discovery of sequence motifs specifically recognised by RBPs [18,19]. Furthermore, various advancements in technologies specifically for the detection of chemical modifications now allow for more precise determinants of modified bases and their associated levels [20,21]. The recent introduction of direct RNA sequencing by Oxford Nanopore Technologies also represents a significant step forward in the accurate detection of a variety of base modifications [22], with the potential for detecting multiple modifications within the same assay and at single-molecule resolution [23,24].

Deep learning, an advanced sub-field of machine learning, has revolutionised the analysis of genomic data in recent years [25]. Deep learning utilises deep neural networks, characterised by multiple layers incorporating potentially millions of ’weights’, allowing well-defined models to learn highly complex patterns and representations from large-scale genomic datasets [26], Figure 1C). To facilitate model development, advancements in deep learning frameworks that efficiently utilise GPU capabilities have significantly enhanced the speed and performance of model training and testing, whilst also expanding the accessibility of deep learning approaches to a broader research community [27,28].

In the context of predicting RNA modification preferences, such as mRNA-RBP interactions, deep learning can serve several purposes (Figure 1D). First, it can address the high levels of noise in experimental datasets, which may be of low resolution or subject to a range of systemic biases [15]. In this regard, deep learning-based predictions can both refine the locations of observed sites and reduce the number of sites called as false positives in data processing pipelines. Second, well-trained models can extend predictions of modified locations to under-explored areas, especially where data is scarce, such as with lowly expressed RNAs, non-coding RNAs, or viral RNA [29,30], potentially leading to significant biological insights. Third, the learned feature spaces from trained models can be leveraged to uncover biologically relevant patterns, such as RNA-binding motifs [31,32], local cis-regulatory elements [33], or secondary structure features. They can also be used to address the impact of sequence variation in silico [34], such as single-nucleotide polymorphisms associated with specific diseases, thereby proving potential phenotypic insights to genetic connections. Additionally, in the context of nanopore direct RNA sequencing, several advanced deep learning approaches have recently been developed to interpret the signals captured as the RNA strand is pulled through a specialised pore, both for accurately determining the base type and modification status for each assayed molecule [23,24].

This perspective provides an overview of recent deep learning-based approaches developed to address where and how modifications select their target sites on mRNAs. Whilst the main focus is on the prediction of mRNA-RBP binding sites, we also highlight publications on other modifications, particularly m${\;\;}^{6}$A and A-to-I editing. This work describes the different types of inputs typically used in training and how various types of layers can be used to connect these inputs to outputs. It discusses how models might be trained in a biologically meaningful way. Additionally, as deep learning methods and applications for RNA modifications are still in their infancy, future prospects in the field are explored, focusing on major challenges and opportunities for applying these techniques in the field.

## 2. Deep Learning for RNA Modifications

In order to decipher the complex regulatory roles of modifications to RNA, including their potential roles in disease, it is essential to be able to accurately pinpoint where and in which contexts these modifications occur. In recent years, numerous deep learning approaches have been published to this end, with some key examples summarised in Table 1. Whilst these approaches can vary greatly, Figure 2 outlines the most typical inputs, features and modelling strategies. First, ’positive’ regions (e.g., known binding locations with some surrounding context) are usually selected for training (Figure 2A). Since deep learning models have a reputation for requiring substantial numbers of training examples, many studies focus on data from large-scale efforts like ENCODE [13,14], or from other well-documented studies in human cell-lines [35,36], often aided by databases such as POSTAR3 [37] or m6A-Atlas [38,39]. Note that many deep learning models depend on proper processing of raw reads as a prerequisite. Rigorous pipelines exist for performing the necessary quality controls, conducting peak-calling and filtering regions also present in paired input samples, thereby generating a list of quality regions, or single-nucleotide positions in the case of some assays such as iCLIP [12], for use in further model training [40,41]. It is important to note, however, that even after peak-calling, certain biases, such as preferences towards certain sequence contexts, may still be present.

To train models to distinguish between affected (e.g., bound by an RBP or methylated) and unaffected states, positive regions are often paired with similar numbers of so-called negative regions. Many recent approaches source these negative regions either from the same gene set to minimise gene selection bias or from general transcriptomic sampling (see [52] for detailed comparison of these two strategies). Some try to address systematic biases by treating regions as negative if they are called as sites bound by other RBPs that are not the RBP of interest [49]. Other methods avoid the need to specifically select negative regions, one example being iM6A [33], which, using an architecture similar to spliceAI for the detection of splice sites [53], aims to detect m${\;\;}^{6}$A sites out from surrounding nucleotides.

### 2.1. Features and Model Architecture

The features to the neural networks can take various forms, but the two which are by far the most prevalent in the prediction of mRNA-RBP binding sites are sequence and RNA secondary structure (Figure 2B). Generally, these types of models are supervised, that is, the ground truth is considered as known and the model is tasked with connecting the supplied input features with these ground truth measurements (e.g., presence or absence of binding) via a set of non-linear functions incorporating weights which are optimised in the training process. For this reason, we focus on supervised approaches here, but other types of models are also extremely useful in biology, such as unsupervised learning of cell clusters from single-cell RNA-sequencing [54] or semi-supervised deep learning for biological imaging analysis [55], just to name two applications.

Due to its simplicity and flexibility, the RNA sequence extracted using the coordinates of the regions surrounding selected positions is frequently processed in a one-hot encoded format, before being presented as input to a neural network (Figure 2B). In this format, each possible base (A,U,C,G) in the RNA sequence is represented as a separate “channel”, which is essentially a vector of the same length as the input sequence with value of one where the sequence encodes for that base and zero otherwise. A popular alternative representation is using *k*-mers, whereby the sequence is broken up into overlapping segments of length *k*, before often being collapsed into a vector of counts for each possible sequence. On the whole, there is a trend towards using wider sequence contexts around positions of interest, with some even including full transcript sequences [33,56], although note that with larger models there may be a trade-off between the maximum input sequence length and the availability of GPU capacity for training. Interestingly, recent deep learning models for RBP appear to show that sequence alone can achieve high performance scores for determining the binding status for a large number of RBPs [29,34,43,57], the locations of m${\;\;}^{6}$A sites [33], and A-to-I editing sites [51].

Figure 2C outlines some popular layer types frequently used in the model architecture of supervised neural networks. Encoded RNA sequence is usually managed using one-dimensional (1D) convolutional layers (termed a convolutional neural network, CNN). Briefly, the model learns a set of weights making up fixed-sized filters, which essentially scan across the input to the layer and assign scores. These scores are subsequently passed via an activation function to the next layer. The initial convolutional layer is especially seen as informative, as it directly connects to the encoded sequence input and can thus be interpreted in terms of de novo motifs or sequence contexts relevant for RBP binding [58]. Whilst CNNs excel at identifying local ’motif-like’ sequence patterns, recurrent neural networks (RNN), specifically those with LSTM (long short-term memory) or GRU (gated recurrent unit) layers, are adept at learning long-term dependencies in sequence data [59,60,61] and bidirectional variations of LSTM can process sequences in both directions, enriching their ability to learn contextual patterns. RNNs have been applied with success in the context of RBP-mRNA interactions, two examples being iDeepE and DeepCLIP [34,57]. In addition, models involving transformer layers implement self-attention mechanisms by assigning variable attention weights to different positions, allowing them to handle all parts of the sequence at once. Transformers have shown promising potential in recently published approaches for RBP-mRNA interaction prediction [44,46,62].

Deep learning models are particularly flexible at combining different layer types, such as LSTM layers following convolutional layers. Fully connected layers, whereby all nodes connect to all nodes in the subsequent layer, are also common, and typically feed into the output layer. Fully connected layers allow for learning highly complex feature spaces, although these feature spaces can be difficult to interpret and these layers should be used sensibly with smaller datasets since they can vastly increase the number of trainable weights in the model. Layers are also interspersed with specialised types of layers such as activation layers, pooling or drop-out layers, which respectively pass features non-linearly between layers, reduce feature dimensions, or limit the number of parameters in the subsequent layer to avoid overfitting [63,64] (Figure 2C). Overfitting occurs when a model achieves a very high performance within the same data on which it is trained, but fails to generalise to new, unseen data, such as new genomic locations. Large models with few examples are especially prone to overfitting, and for this reason, it is important to assess model performance on only unseen data (see below). Overall, due to wide possibilities for complex arrangements and parameterisations, different models with similar feature sets can potentially perform very differently. For this reason, it is important that approaches are carefully optimised for the given problem and cannot be treated as ’black-box’ approaches for machine learning.

### 2.2. Incorporating RNA Secondary Structure

Since the RNA-binding domains of RBPs vary in their ability to recognise and bind RNA secondary structures, RNA conformation can play an important determining role in the prediction of mRNA-RBP interactions and is therefore frequently considered as an input feature in models. GraphProt, based on support vector machines (SVMs), was one of the first models to extensively incorporate predicted secondary structure information, encoded as graph kernels [65] and has since been superseded by deep learning-based methods. Many of these models encode secondary structure features into a graph, where each node represents a nucleotide in the sequence and edges symbolise their interactions [47,66,67]. This graph representation is processed via a graph convolutional network (GCN) (Figure 2C), which applies a series of convolution operations, aggregating features from neighbouring nodes by taking into account both its individual features and the structure of its immediate surroundings in the graph.

The majority of these models focus on predicted RNA secondary structures via the application of computational tools (e.g., RNAfold or RNAshapes [68,69]. These tools work by calculating the minimum free energy (MFE) structure from a given sequence based on sophisticated dynamic programming algorithms. As RNA structure can be stochastic within cells, one approach found it advantageous to consider base-pairing probabilities instead of a single MFE configuration [70]. Alternatively, PrismNet accommodates experimentally determined in-vivo secondary structures by utilising the experimental icSHAPE method [30,44], which provides a score at genomic positions representing double-strandedness or single-strandedness of the transcribed RNA [71]. Notably, these structures appear to remain stable across different cell types [72], yet their observed variations seem able to decipher dynamic, tissue-dependent mRNA-RBP binding [43]. A more recent model, HDRNet, takes this concept a step further by testing the capabilities of in-vivo secondary structures to predict dynamic RBP binding in a given cell context using a model which was trained on another context, with promising results [44]. In addition, the same study further supported the advantage of using in vivo structures over computational predictions, finding that models using in vivo structures always outperformed their counterparts based on RNAfold-predicted structures in place of the icSHAPE scores.

However, note that the inclusion of the RNA secondary structure and sequence in parallel does not always guarantee an improvement in performance over sequence alone, and is instead likely to be dependent on the underlying biology of the protein under study [57]. Interestingly, for certain RBPs such as PABPC4, METAP2, DDX55, and DGCR8, performance was actually found to be higher for a model using only structure-based features compared to sequence-only features [30]. However, the same study did show that, on average across all tested proteins, a combination of in vivo structure and sequence resulted in the best performance (an AUROC of 0.850 compared to 0.797 and 0.758 for sequence-only and structure-only models, respectively, where an AUROC of 1 implies a perfect model; see below for description of measures of model performance). Since performance variations are likely reflecting the underlying biology of the protein under study, a closer look into these statistics may provide useful clues of properties of binding behaviours of lesser known RBP groups. In addition, note that the role of secondary structure via deep learning for the prediction of other modifications such as RNA-editing and m${\;\;}^{6}$A remains less explored. Whilst the ADAR proteins catalysing RNA-editing are known to have strong preferences for double-stranded RNA, m${\;\;}^{6}$A sites typically appear associated with single-stranded RNA, although the precise nature of the relationship between m${\;\;}^{6}$A and RNA structure does remain somewhat unclear [73].

Finally, note that models are not limited by only sequence and/or secondary structure features, but can flexibly be extended to incorporate further features: for example, RNAprot has shown benefits in including additional features such as sequence conservation [48], and, given RBPs have preferences for specific genomic features, including information of the locality of the bound region (e.g., 5${\;\;}^{\prime }$ UTR, CDS, 3${\;\;}^{\prime }$ UTR) also seems to help predictions [49]. Moreover, progress in the use of large language models for genomic sequence, such as bidirectional encoder representations from transformers (BERT), has accelerated recently [42] and appears advantageous for the prediction of RNA-RBP interactions [44,46]. For example, HDRNet adapts BERT to encode sequences of interest broken into short *k-mer* stretches as dynamic representations that appear efficient at capturing both local contexts and long-range dependencies, reflected in the impressive performance of their models [44].

### 2.3. Perspective on Current Models

Due to the limited number of large data resources over which modelling can be applied systematically on a broad scale, a number of the RBP models mentioned in Table 1 focus on the same datasets (e.g., ENCODE [14] for RBPs in the HepG2 and K562 cell lines). Despite this, the overall approaches employed by the various models are often difficult to compare due to differences and/or ambiguities in model setups, such as variations in the choice of negative sets, region sizes, and chosen strategy for dividing examples between sets used for training and testing sets used for assessing model performance (for example, holding out individual positions vs. whole genes or chromosomes, see below for further discussion on this aspect) (see Table 1 for set-ups of individual models). On the other hand, despite the different modelling approaches, the performance of a given model may be more influenced by the characteristics of the protein under study than by specific model architectures or feature choices. For instance, AGO2, involved in RNA-binding functions via miRNAs [74], and ALKBH5, an eraser of N6-methyladenosine, often receive lower performance scores relative to other RBPs [34,67]. This could be attributed to the low dependency of that protein on only sequence-based features, although factors such as poor data quality or insufficient numbers of training sites to work with likely also play a significant role.

Moreover, it should be noted that trained models are not immune to biases present in the original data. For example, one study noted an inverse correlation between AUC performance and GC-richness in the RBP’s sequence preference [34]. Efforts to mitigate sequence biases in immunoprecipitation-based data are ongoing [29,67,70]. For example, a promising recent method, RBPnet, integrates matched input signal in a mixture-model approach, allowing the neural network to separate ‘true’ signal from systematic biases that can be inferred from the input [29]. Another method, HPNet, attempts to mitigate the influence of systematic nucleotide bias in the data by employing ’context-averaging’ [67]. Whilst these attempts are important for improving the quality of binding site predictions, it should be noted that the field’s heavy reliance on eCLIP data underscores the need for further attention to this issue [14]. Incorporating orthogonal experimental information, such as data from the RNA-editor methods [16,17], could lead to more confident results, although such approaches have to date not yet been tested.

In addition to the above considerations, overfitting remains a significant challenge, caused by the scarcity of reliable input data and labels for training, particularly to RBPs with few binding sites, or rarer chemical modifications. One promising option to mitigate this is via transfer learning, whereby one trains on a broad set of modifications and then fine-tunes on individual modifications [75]. The use of recent foundation models for genomic sequence can help in this regard [42,76]. These large models are pre-trained over a broad range of genomes, and sequence embedding vectors can be extracted or models fine-tuned for use in specific problems, such as mRNA-RBP site prediction, thus aiding in circumventing issues with low dataset sizes [46].

### 2.4. Model Performance and Choice of Background Set: The Hunt for Biologically Relevant Results

It should be noted that due to competing approaches on similar data, there is often a pressure to demonstrate the best model performance in published works. Model performance can be measured in multiple ways. The most typical scores are accuracy (the proportion of correct predictions) and the area under the curve (AUC), which balances the sensitivity of the model to find true sites and the specificity to exclude those which are not true sites, with 1 being a perfect model and 0.5 suggesting the model guesses randomly. Alternatively, the area under the precision recall curve (AUPRC) balances precision, which is the proportion those sites predicted which are actually modified, and recall, which is the proportion of modified sites that are also predicted as such. AUPRC ranges between 0 and 1 and is usually preferred over AUC in situations where numbers of positives and negatives are unbalanced, as the AUC can be highly misleading in these situations. In any case, it is normally preferable to quote a range of statistics in order to comprehensibly describe and compare model performance.

Whilst one generally would expect to trust predictions derived from a well-performing model over a lesser-performing one, it is important to realise that there are situations where a well set-up model can show poorer performance according to these above metrics, yet yield more biologically relevant results. One example is the use of stringent background sets, which may lead to poorer performance metrics but actually ensures that the learned feature space aligns with the biological problem and avoids capturing irrelevant information. To address the impact of choice of the negative set, a recent study systematically benchmarked a range of models in the context of RBP binding [52]. They considered two different background sets: one based on random sampling within the same genes as the positive positions, and one based on sampled positions that were experimentally-defined binding sites of other RBPs. The rationale of the second set was that by sampling positions of other RBPs to the one of interest, one can avoid learning features associated with potential experimental biases between the positive and negative set. Indeed, they found that performance dropped for this second set, suggesting that a portion of the measured performance of the less strict background may be capturing over-representation of experimental bias in positive sets rather than true biological signal.

### 2.5. Further Considerations for Modelling Approach

Figure 3 illustrates some suggestions for designing and training deep learning models to maximise biological insight. First, as mentioned, careful selection of negative site location is essential (Figure 3A). In addition, RBPs are not uniformly distributed across transcripts, but are often localised to highly specific regions, such as near splice sites or in the 3${\;\;}^{\prime }$ UTR [14], and m${\;\;}^{6}$A tends to be enriched near the stop codon. Since sequence and/or secondary structure preferences are highly variable across transcripts, general sequence patterns relevant to the transcript region or the presence of these specific features might be over-represented in intervals around the positive sites. Similar to the above situation with experimental biases, this could result in the model appearing to perform well, but in a way that is capturing region preferences independent of what is relevant to the binding of the specific modification. To circumvent this, when constructing machine learning models, it is suggested to consider carefully matching background sites according to transcript features, and preferably on genes that display similar expression distributions and/or are targets for the RBP of interest [77].

Second, when training models, it is common practice to employ cross-fold validation (Figure 3B,C). Here, one allocates positions to a number of folds, typically 5 or 10, and trains the model on all the folds but one, using the held-out fold to assess model performance. Producing predictions for each held-out fold allows one to gain an overall picture of model performance, and since performance statistics are only calculated on sets not including within training, these statistics should not be inflated by potential overfitting of the model. Frequently, in order to allocate training examples to different folds, sites are simply randomised across these five folds, this being the case for a large number of the models highlighted above (see Table 1 for strategies employed for allocating folds by selected recent mRNA-RBP models). However, in the case of post-transcriptional modifications, there is a strong argument for exercising caution, since one single transcript region can harbour multiple modified sites, forming clusters. This means that highly overlapping input features (e.g., sequence or secondary structure) will occur across multiple different folds, potentially leading to inflated performance statistics. Thus, it is highly recommended to allocate folds in such a way that any given gene is only found within a single fold (Figure 3B). Many published deep learning models within the genomics field take this a step further, by holding out entire chromosomes in order to strictly ensure that there is no leakage between the training and testing sets, with one example from Table 1 being RBPnet [29]. In addition, SpliceAI takes an even stricter approach by holding back paralogs from the training set in order to addresses the issue of sequence similarity due to common gene ancestors [53], and it would be interesting to see how this approach affects the prediction scores of current mRNA-RBP interaction modelling approaches. Additionally, along similar lines of argument, when interpreting a model, it is also recommended to work with prediction scores or effects of in silico mutation using a model whereby the involved inputs were never seen within the training set (Figure 3C).

In conclusion, it is important for focus to shift towards biological interpretations beyond mere performance statistics when assessing a model. Lower performance scores might indicate that the problem is set up more stringently, controlling for more confounding variables. These models may have greater potential to provide interesting biological insights, and further efforts are required to fully explore these aspects.

## 3. Some Major Future Perspectives

Deep learning within genomics, including the field of RNA modifications and particularly the context of mRNA-RBP interactions, is gaining a lot of traction in recent years, with a lot of recently published approaches (not limited to those presented in Table 1). However, there are still a number of distinct challenges and opportunities to be taken into account, so that the field can move further forward in terms of biological discoveries. Four are briefly discussed below.

### 3.1. Generalisability across Cell Types and Species

Current experimental methods for studying RNA-binding proteins (RBPs) typically require substantial input material, leading to a heavy reliance on cell lines. This reliance has resulted in data skewed toward a group of specific human cell lines, most notably K562, HepG2 and HEK293T [14]. Whilst the sequence itself remains consistent across different cell types, variable expression patterns influence binding opportunities, which are likely further driven by distinct regulatory ’grammar rules’ driving observed cell-type-specific patterns [78]. Furthermore, secondary structure differences have been shown to significantly affect RBP binding disparities between K562 and HepG2 cells [43], which can be leveraged for predicting dynamic binding patterns across cell types [44]. Without experimental data covering a broader range of cell types, the relevance of such differences cannot be thoroughly explored. For example, RBPs like TDP-43 and FUS are implicated in memory formation through the creation of sub-cellular RnP granules; such context-specific roles are not adequately represented in the most commonly assayed cell types [79].

Moreover, the field is heavily biased towards human data, with a notable lack of data from other species. In plants, for example, RBPs play essential roles in growth, development, and stress response, yet high-quality RBP binding data for these species are scarce [80], although recent improvements in immunoprecipitation and RNA-editor approaches aimed specifically at plants are now boosting this area [77,81]. The POPSTAR3 database, which compiles RBP binding data across seven species [37], further highlights the limited share of non-human data available. Expanding RBP binding data in non-human species is crucial not only for the understanding modification-associated regulatory mechanisms in each specific species, but also for deciphering evolutionary relationships in RBP binding. Furthermore, note that a significant challenge with non-human species is the limited understanding of which proteins possess mRNA binding capabilities. For this reason, the integration of deep learning approaches aimed at predicting RBPs and their binding domains [82,83,84,85] and their subcellular locations [86] could be key to advancing experimental efforts across various species.

### 3.2. Focus on Model Interpretation

Interpreting deep learning models, given their highly non-linear feature spaces involving large numbers of weights, is challenging yet crucial for understanding biological contexts [26,87]. Reassuringly, there is a growing trend towards models considering what sequence motifs and/or secondary structure contexts might be driving RBP binding to RNA. In particular, in silico mutagenesis or related approaches can be used to interrogate how base changes in the input sequences influences the binding predictions, and are further useful for informing motif detection algorithms [88,89,90,91]. As an example, Grønning et al. used their models to show that point mutations known to cause exon skipping were predicted to result in increased binding of the RBP SRSF1, which has known roles in exon inclusion [34], and the authors of iM6A looked at the impact of single nucleotide variants on m${\;\;}^{6}$A deposition probabilities [33], which showed agreement with experimental data. Moreover, two recent studies leverage their models to make predictions on viral RNAs, where there are very few training data to work with [29,30], and the results were at least in part validated by external datasets. Significant challenges persist, however, as RBPs can be highly redundant and have highly redundant binding sites, such that a single-nucleotide mutation may not be sufficient to alter the binding probability. Therefore, more work is required to build robust frameworks of how sequence variation affects molecular function and disease through its impact on RBP binding.

### 3.3. Extensions to Predictions on Non-Coding RNAs

Non-coding RNAs, such as long non-coding RNAs, enhancer RNAs, microRNAs, and others, exhibit highly diverse functional roles and pronounced cell-specificity [92]. Moreover, the functional roles of these RNA species in the context of their interactions with RNA modifiers such as RBPs or m${\;\;}^{6}$A remain poorly understood [93]. One suggested function of lncRNAs is to act as molecular scaffolds or decoys, potentially sequestering RBPs from target genes, with implications in immune regulation [94]. Additionally, recent studies have revealed an enrichment of m${\;\;}^{6}$A modification in non-coding RNAs, with m${\;\;}^{6}$A-reader RBP YTHDC1 playing a role in maintaining RNA integrity [95].

On the whole, the limited availability of data on non-coding RNA modifications poses a significant challenge in terms of training deep learning models. One approach could be to employ models trained on mRNA-based data to predict modifications in non-coding RNAs. Due to the current lack of experimental methods, however, validations of these observations remain difficult; therefore, technological advancements that can lead to high quality and throughput at these regions could have both large impact in terms of validating current deep learning based observations, as well as training new models.

### 3.4. Cooperative Contexts and Interplay with Other Modifications

RBPs do not operate in isolation, but often exhibit redundant behaviour or act in collaboration or competition with other RBPs. For instance, the YTH-domain m${\;\;}^{6}$A reader RBPs display highly redundant functions [96,97], while the RBPs HuR and AUF1 are known to compete for binding sites, affecting the stability of shared mRNA targets [98]. Moreover, different types of modifications do not act independently. For example, the presence or absence of A-so-I RNA editing can significantly modify RBP binding patterns [99,100]. Such interactions underscore the highly complex nature of RNA regulation, where modifications and RBPs form a dynamic dependency network. Therefore, studying the binding patterns of a single RBP or modification may not sufficiently capture the nuances of RNA stability and decay. Indeed, research in this area has demonstrated that considering the full repertoire of RBPs yields a more accurate prediction of RNA half-life than analysing any individual RBP [43]. However, a significant limitation in current research is the scarcity of experimental methods capable of establishing potential cooperative binding locations on a genome-wide scale; addressing this gap would provide essential data for establishing ground truths on which to train deep learning-based modelling frameworks.

## 4. Conclusions

This article has provided an overview of recent advances in deep learning in the context of RNA modifications, highlighting areas in which there are distinct challenges and opportunities. It is important to emphasise that this is a cyclical process, with experimental data forming a basis for deep learning models, and these improved models, in turn, can guide the development of either improved or more targeted experimental methodologies. Consequently, future collaboration between experimental and computational biologists will be key for driving progress in the RNA modification field, allowing for the construction of powerful and highly interpretable models able to answer biological questions in a range of species and contexts.

## Figures and Tables

**Figure 1 genes-15-00629-f001:**
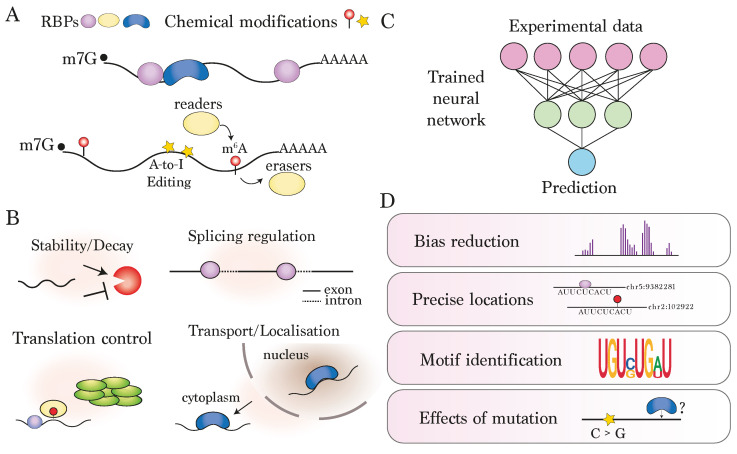
**Introduction to deep learning for the prediction of modified sites on RNA**: (**A**) RNAs are modified co- or post-transcriptionally by RNA-binding proteins (RBPs) and a range of chemical modifications. The study of these modifications transcriptome-wide is collectively termed epitranscriptomics. (**B**) Example roles of RNA modifications played by RNA-binding proteins and/or m${\;\;}^{6}$A methylation. (**C**) Schematic depicting neural network, which typically starts with transcriptome-wide measurements of modified positions and makes some prediction based on the trained model. (**D**) Motivations for using deep learning approaches, including handling of bias, identification of precise binding locations, or model interpretation in terms of motif identification or assessing effects of sequence mutation.

**Figure 2 genes-15-00629-f002:**
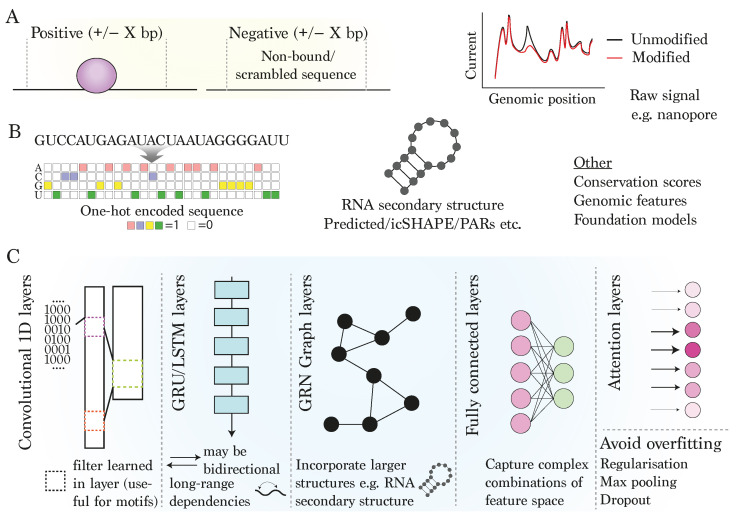
**Strategies and outcomes for modelling RBP binding using deep learning techniques**: (**A**) definition of positive and negative sites and inclusion of surrounding context. (**B**) Inputs: typical positive training examples include sequence and/or RNA secondary structure at and surrounding locations of known binding sites, together with negative regions of equal size without detected binding. (**C**) Modelling: Popular deep learning-based layers include single-dimensional (1D) convolutional layers for detecting local sequence motifs from the input data; gated recurrent unit (GRU) or long short-term memory (LSTM) layers for capturing long-range dependencies impacting RBP binding; graph (GRN) layers for capturing higher-order interactions between RNA nucleotides/structures; a fully connected layer, allowing the model to learn complex patterns; and attention, which focuses on important parts of the input data.

**Figure 3 genes-15-00629-f003:**
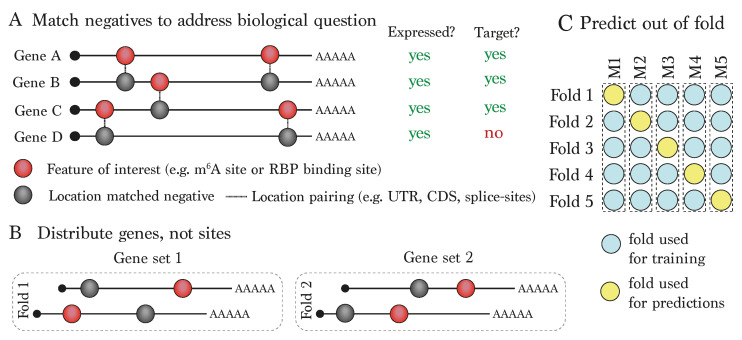
**Recommendations for deep learning model training and prediction:** (**A**) Choice of negative (’non-modified’) locations can have a large influence on the biological applicability of the model. Suggested options could be to match modified positives according to feature location, expression and/or target status. (**B**) When performing cross-validation, it is recommended to avoid information leakage by distributing genes across the folds rather than individual sites, whose sequence could overlap a site in another fold. (**C**) In order to honestly assess the model, it is recommended to predict out of fold.

**Table 1 genes-15-00629-t001:** Examples of some recent models for RNA modifications with emphasis on RNA-binding protein (RBP) binding. Note that this is not meant as an exhaustive list, but to cover a variety of currently available models. Abbreviations: RNASS: RNA secondary structure; CNN: convolutional neural network; RNN: recurrent neural network; SVM: support vector machine; GCN: graph convolutional network; MIL: multiple instance learning. For more detailed specifics of each model, please see cited references.

Name	Data	Description	Model	Ref
HDRNet	RBP	Sequence + in vivo RNASS (icSHAPE) + DNABERT [42], data shared with PrismNet [43], 101 bp regions, random assignment of positions in test/training.	Attention	[44]
BERT-RBP	RBP	Sequence + RNASS + DNABERT [42], 101 bp regions, data as per RBPsuite [45], random assignment of positions in test/training.	Attention	[46]
RBPnet	RBP	Sequence to signal mixture approach for bias correction, 300 bp windows, chromosome-wise splits to test/training.	CNN	[29]
DeepPN	RBP	Sequence + RNASS, bound-genes sourced negatives, 501 bp regions, random assignment of positions in test/training.	CNN/GCN	[47]
PrismNet	RBP	Sequence + in vivo RNASS, 101 bp regions, negatives with >40% icSHAPE coverage sampled from transcriptome, random assignment of positions in test/training.	CNN/attention	[43]
RNAProt	RBP	Multiple variable features, inc. sequence, RNASS, conservation, etc., 81 bp regions, random assignment of positions in test/training.	RNN	[48]
DeepCLIP	RBP	Sequence, matched-gene negatives for training, up to 75 bp regions, random assignment of positions in test/training.	CNN/RNN	[34]
DeepRiPe	RBP	Multitask models covering 59 RBPs, Sequence (150 bp regions derived from 50 bp bins) and genomic feature information (250 bp regions), random assignment of bins in test/training.	multi-output CNN	[49]
iM6A	m${\;\;}^{6}$A	Sequence-based m${\;\;}^{6}$A site prediction, surrounding unmethylated sites as negatives, human+mouse	CNN	[33]
m6Anet	m${\;\;}^{6}$A	Trained on nanopore signal for molecule-resolution m${\;\;}^{6}$A prediction.	NN/MIL	[23]
EditPredict	A-to-I	Sequence, predict A-to-I editing sites sourced from REDIportal [50], non-edited sites as negatives, up to 200 bp regions, multi-species.	CNN	[51]

## Data Availability

The original contributions presented in the study are included in the article, further inquiries can be directed to the corresponding author.

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
