# Peer review of "Deep Learning for Elucidating Modifications to RNA—Status and Challenges Ahead"

_genes, 2024, doi:10.3390/genes15050629_

Round 1
Reviewer 1 Report
Comments and Suggestions for Authors
In this manuscript, Rennie present a timely perspective on integrating deep machine learning into studying the post-transcriptional modifications of RNAs. It described the up-to-date development of computational methods and bioinformatic tools and databases in mapping RNA modifications. Moreover, it offered insights into taking the challenges in methodologies and techniques development.
I would suggest that Rennie could include due part of description and discussion on use of deep learning in studying modifications of tRNAs. Moreover, a discussion of use of deep learning in studying the effect of three dimension structure on the modification of RNAs are recommended.
Comments on the Quality of English LanguageMinor proofreading is desirable.
Author Response
In this manuscript, Rennie present a timely perspective on integrating deep machine learning into studying the post-transcriptional modifications of RNAs. It described the up-to-date development of computational methods and bioinformatic tools and databases in mapping RNA modifications. Moreover, it offered insights into taking the challenges in methodologies and techniques development.
I would like to thank the reviewer for their time and for their positive comments relating to my submitted manuscript. I would also like to thank the reviewer for their comments below:
I would suggest that Rennie could include due part of description and discussion on use of deep learning in studying modifications of tRNAs.
Since large scale deep learning effects for predicting RBP binding/ internal chemical modifications have so far focussed on mRNA (due to limitations in experimental procedures, and the need for large numbers of training examples in modelling), I have clarified in the article that the text discusses models based on modifications on mRNAs, although it is highlighted that more work does need to be done with regarding to other types of RNAs.
Moreover, a discussion of use of deep learning in studying the effect of three dimension structure on the modification of RNAs are recommended.
I have now greatly expanded on the use of RNA structure in the deep learning models used to predict RBP binding, including more about the relative contributions of sequence and structure in determining model performance and more discussion about use of in-vivo vs in-vitro structures as an input into the trained models.
Reviewer 2 Report
Comments and Suggestions for Authors
Overall it is a nice review on the use of deep learning to detect modifications of RNA, however, I do find there are some peculiar attentions to detail in some methods/sections and not in others.
For example, the abstract could have much more references to what specifically is going to be reviewed in the review.
Another example is line 115, for "one-hot encoded format", what does this mean?
I love the clarity Figure 2C brings!
I think Table 1 needs to come much sooner in this manuscript for coherence purposes. it would also be nice if this table had another column for number of RBPs found in a given model. (Or an Upset plot to show overlap?)
Is there a quantitative or estimate you can give for the contribution of sequence and secondary RNA structure for RNA modifications? Maybe even based on current modeling efforts?
Comments on the Quality of English LanguageFor a review paper, I do expect near perfect writing, so the presence of minor typos and english errors seems more significiant. Nonetheless, I could understand the major take homes of what was written.
(some lines where there are minor edits: line 20-21, lines 30-32, Table 1 caption)
Author Response
Overall it is a nice review on the use of deep learning to detect modifications of RNA, however, I do find there are some peculiar attentions to detail in some methods/sections and not in others.
I would like to begin by thanking the reviewer for their time and positive comments relating to my submitted manuscript.
For example, the abstract could have much more references to what specifically is going to be reviewed in the review.
This is a very fair point and I have expanded the abstract to give more details.
Another example is line 115, for "one-hot encoded format", what does this mean?
I have now expanded on the description of one-hot encoding, both in the text and the figure. I have also looked at other terms such as overfitting, training/test and improving the clarity of the explaining of the model performance statistics.
I love the clarity Figure 2C brings!
I think Table 1 needs to come much sooner in this manuscript for coherence purposes. it would also be nice if this table had another column for number of RBPs found in a given model. (Or an Upset plot to show overlap?)
According to the suggestion, I have now moved the table much further up in the text. I have added a column clarifying what modification is under study and, as I believe the overall approach to be more interesting than the number of modelled RBPs considered in each individual study, I have chosen to greatly expand on the details of each of the selected approaches in the table, including in some cases pointing out if the same datasets are used.
Is there a quantitative or estimate you can give for the contribution of sequence and secondary RNA structure for RNA modifications? Maybe even based on current modeling efforts?
I thank the reviewer for the suggestion. I have expanded on the section for RNA secondary structure overall and included information about the relative contributions of structure and sequence on the prediction of RNA binding proteins.